

# Restoring South African subtropical succulent thicket using *Portulacaria afra*: root growth of cuttings differs depending on the harvest site during a drought

Alastair J. Potts[1], Robbert Duker[2], Kristen L. Hunt[1], Anize Tempel[1] and Nicholas C. Galuszynski[1]

[1] Spekboom Restoration Research Group, Department of Botany, Nelson Mandela University, Gqeberha, Eastern Cape, South Africa
[2] C4 EcoSolutions (Pty) Ltd, Cape Town, South Africa

## ABSTRACT

The restoration of succulent thicket (the semi-arid components of the Albany Subtropical Thicket biome endemic to South Africa) has largely focused on the reintroduction of *Portulacaria afra* L. Jacq—a leaf- and stem-succulent shrub—through the planting of unrooted cuttings directly into field sites. However, there has been inconsistent establishment and survival rates, with low rates potentially due to a range of factors (*e.g.*, post-planting drought, frost or herbivory), including the poor condition of source material used. Here we test the effect of parent-plant and harvesting site on the root development of *P. afra* cuttings in a common garden experiment. Ten sites were selected along a ∼110 km transect, with cuttings harvested from five parent-plants per site. Leaf moisture content was determined for each parent-plant at the time of harvesting as a proxy for plant condition. Root development—percentage of rooted cuttings and mean root dry weight—was recorded for a subset of cuttings from each parent-plant after 35, 42, 48, 56, and 103 days after planting in a common garden setting. We found evidence for cutting root development (rooting percentage and root dry mass) to be strongly associated with harvesting site across all sampling days ($p < 0.005$ for all tests). These differences are likely a consequence of underlying physiological factors; this was supported by the significant but weak correlation ($r^2 = 0.10$–$0.26$) between the leaf moisture content of the parent-plant (at the time of harvesting) and dry root mass of the cuttings (at each of the sampling days). Our findings demonstrate that varying plant condition across sites can significantly influence root development during dry phases (*i.e.*, intra- and inter-annual droughts) and that this may be a critical component that needs to be understood as part of any restoration programme. Further work is required to identify the environmental conditions that promote or impede root development in *P. afra* cuttings.

Corresponding author
Alastair J. Potts, potts.a@gmail.com

## INTRODUCTION

Arid and semi-arid (dryland) ecosystems are characterised by low and, in certain systems, unpredictable rainfall, which can cause considerable stress to plants. The introduction of additional, prolonged and compounding stressors, such as non-native herbivores, to these environments often compromises ecosystem functioning and triggers a transition to an alternate, degraded state (*Verwijmeren et al., 2013*; *Scheffer et al., 2001*). Restoration of these drylands requires the reversal of these transitions and, ultimately, the reintroduction of the ecosystem processes that have been lost (*James et al., 2013*). This study focuses on the succulent component of the Albany Subtropical Thicket biome (*i.e.,* arid and valley thicket types *sensu Vlok, Euston-Brown & Cowling (2003)*), which occurs in the semi-arid southern coastal lowlands of South Africa, and has been extensively degraded with large-scale restoration initiatives underway (*Mills et al., 2015*; *Mills et al., 2023*).

The degradation of succulent thicket serves as a notable example of the transition between states in a dryland ecosystem. By 2002, up to 80% of the ecosystem had been degraded to some extent (*Lloyd, van den Berg & Palmer, 2002*). The unsustainable browsing of livestock within these succulent thickets has resulted in the transition from a dense, impenetrable closed-canopy system (typically dominated by *Portulacaria afra* Jacq., a succulent shrub {sometimes tree} endemic to southern Africa) to an open pseudo-savanna. This degraded habitat is typically characterised by an open matrix of bare soil, dwarf shrubs, and ephemeral, and largely ruderal, herbs and grasses, with a few remnant canopy-dominant tree species, such as *Pappea capensis* Eckl. & Zeyh. (*Lechmere-Oertel et al., 2008*). This loss of plant cover, including *P. afra*, results in the disruption of various ecological processes. Exposed soils are especially prone to: erosion due to higher rates of water runoff (*Mills & de Wet, 2019*; *van Luijk et al., 2013*; *Cowling & Mills, 2011*); loss of soil organic carbon (*Mills & de Wet, 2019*; *Lechmere-Oertel et al., 2008*); and a disruption of soil microbial communities (*Schagen et al., 2021*). Furthermore, the loss of the cool, damp understory microclimate required for germination (*Wilman et al., 2014*; *Sigwela et al., 2009*) halts woody species recruitment in the degraded landscape (*Lechmere-Oertel, Cowling & Kerley, 2005*).

Due to the intrinsic drought-hardy nature of *P. afra* (*Guralnick & Ting, 1987*) and its ability to readily grow vegetatively from cuttings, the restoration of succulent thicket has focused predominantly on the active planting of unrooted *P. afra* cuttings (with limited guidance on selecting source material to optimize root development and establishment). Where successful, this practice has regenerated soil organic carbon (*Mills & Cowling, 2006*), promoted the return of microbial communities (*Schagen et al., 2021*), and facilitated the return of natural recruitment dynamics (*Galuszynski, 2023*; *van der Vyver et al., 2013*). However, this restoration has failed to produce consistent results, with large-scale planting initiatives in the Great Fish River Nature Reserve and Addo Elephant National Park (representing ~21.3 million planted *P. afra* cuttings) reporting a mean survival of 28% with high variation across sites (*Mills & Robson, 2017*). These low establishment rates may be a consequence of a host of (potentially interacting) factors, such as local soil properties (*Mills et al., 2011*), moisture availability (*Galuszynski et al., 2023*), landscape position effects (*Duker et al., 2020*), and/or herbivory (*van der Vyver et al., 2021*). However, all the cuttings

used in these plantings were sourced from local stands of *P. afra*, likely encompassing a wide range of spatial and temporal environmental conditions. This approach did not consider the potential influence of parent-plant condition on the outcome of restoration initiatives. The condition of parent-plants at the time of harvesting may significantly influence cutting establishment rates and survival; here we explore the variation in rooting success amongst parent-plants and sites.

Restoration programs are often characterised by low success rates, with a global estimated average of 52% survival (reviewed by *Godefroid et al., 2011*), which is notably higher than that reported in the *P. afra* reintroduction programs. Low survival has been elsewhere attributed to the poor condition of the plant material used in planting programs (*Duguma et al., 2020*; *Kildisheva et al., 2017*), resulting in propagules that are unlikely to tolerate local environmental stresses (*Godefroid et al., 2011*). In restoration programs around the world, the practice of sourcing propagation material from healthy parent individuals (*Kildisheva et al., 2017*; *Amri et al., 2010*; *Husen & Pal, 2007*; *Chalupa, 1993*) and populations is recommended (reviewed in *Bucharova et al., 2017*; *Houde, Garner & Neff, 2015*; *van Andel, 1998*). However, this practice is not currently taken into consideration when utilising *P. afra* for restoration purposes.

Succulent thicket spans a diverse range of environmental conditions, across various drainage basins, elevations, soil types, and differences in the timing and amount of rainfall received (*Mills et al., 2011*; *Vlok, Euston-Brown & Cowling, 2003*). Rainfall in this system is highly variable at regional and local scales, including frequent and often prolonged droughts that continue for many years, and may include months with little to no rainfall (*Archer et al., 2022*; *Mahlalela et al., 2020*; *Palmer et al., 2002*). We suspect that the physiological condition of *P. afra* at the time of harvesting, which is dependent on the prevailing conditions (*Bews & Vanderplank, 1930*), may strongly contribute to the likelihood of establishment and subsequent survival of planted cuttings. Parent-plant condition may have contributed towards the highly variable survival in large-scale plantings (*e.g.*, 0–100% reported in *van der Vyver et al., 2021*).

To improve our understanding of the potential effects of parent-plant condition on succulent thicket restoration success, this work quantifies root development of cuttings grown in a common garden with the cuttings harvested (on the same day) from parent-plants at ten different sites across an environmental gradient (a transect of ~110 km). We consider the rate of root development—measured here using two metrics: the percentage of rooted cuttings and mean root dry weight—as a crucial determinant of cutting establishment and survival, as sufficient rooting is necessary for moisture absorption during the often short periods of water availability (*van Luijk et al., 2013*) and water storage during the regular long period between rain events (*Eggli & Nyffeler, 2009*). These sites cover a wide array of environmental conditions, from various elevations (190 to 635 m), to a climate spectrum that ranges from the drier, inland northern areas, to the more moist, near-coastal southern environments. Our results demonstrate that there are significant differences in root development across the various harvesting sites when grown in a common garden, which we attribute to differences in parent-plant condition. These results should serve as important guidance to future restoration programs.

## METHODS

### Recent climatic conditions

The Eastern Cape of South Africa experienced a severe and prolonged drought starting in 2015 (*Archer et al., 2022*; *Mahlalela et al., 2020*), and continued up to the sampling period of this experiment (October 2021, Fig. 1, Fig. S1, Table S1). This included below-average rainfall across all peak rainfall periods over much of the Eastern Cape province, including some of the driest winters recorded since 1981 in some areas along the western interior of the region (*Archer et al., 2022*) (Fig. 1). Seasonal greenness (used as a proxy for plant productivity and condition) was lower than pre-drought conditions for winter and summer (*Archer et al., 2022*), suggesting that plant condition was generally poor across the region, including along the transect along which the *P. afra* was harvested for this study. However, regional patterns are a poor reflection of localised vegetation condition. Thus, to characterise the drought within the study area and to contextualise the climate differences between harvesting sites, monthly rainfall estimates were extracted from the *CHIRPS* dataset—an extrapolated rainfall dataset spanning 1981 until the present at a 0.05 arc-degrees resolution generated and hosted by the Climate Hazards Center—at the various sites sampled along the transect (described above). This demonstrated that rainfall was below the monthly average for 14 of the 17 months prior to harvesting of the plant material used in the experiment (Fig. 1, Fig. S1, Table S1). In addition, in the preceding eight months, the 12-month standardized precipitation index (spi) was lower than $-1$, shifting between moderately to severely dry conditions. The precipitation data was extracted from the *CHIRPS* archive (*Funk et al., 2015*) using the library *chirps* ver 0.1.4 in R (version 4.2.2; *R Core Team, 2023*). The spi index was calculated using *SDI()* function of the *drought* library (ver 1.1; *Hao & Yu, 2022*).

### Site selection and harvesting from parent-plants

Ten sites were sampled along a transect spanning ∼110 km in a north-east to south-west orientation (Fig. 2A), representing three distinct positions in the landscape: inland mountains (sites 1, 2, 5, and 6), inland lowlands (sites 3, 4, 7, and 8) and coastal hills (sites 9 and 10). The rainfall gradient found along the transect is steep as a result of orographically-induced precipitation driven by the Cape mountain chains combined with strong rain shadow effects within the extended flat basins between the mountains (Fig. 2, Fig. S1, Table S1). Thus, sites in the inland mountains received more rain than neighbouring sites located in the extended rain shadow of the inland lowlands, whereas sites in the coastal hills received the highest rainfall due to low continentality.

Although there are notable morphological differences across the distribution of *P. afra* (*Van Jaarsveld & Le Roux, 2021*), no obvious differences were observed in leaf size or plant shape across the transect, which spans a historically continuous distribution of succulent thicket (*i.e.,* we do not consider these to be distinct or isolated populations). Five individuals were selected from each site, and 42 cuttings were harvested from each individual; note that site 5 only had four individuals (due to the loss of material during transport). Plants were selected from areas in the landscape that were considered free from herbivores, *e.g.,* within the road reserve (but in an elevated position unaffected by the road) or within fenced-off

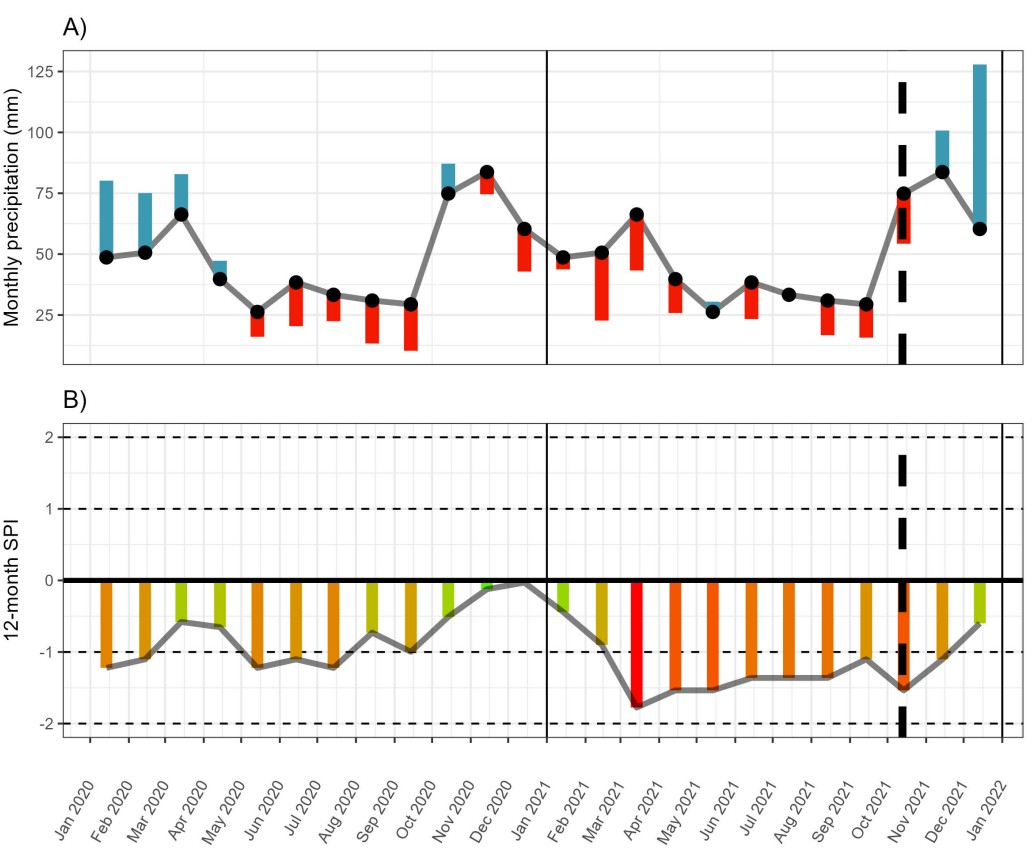

**Figure 1 Monthly rainfall and standardised precipitation index.** (A) The monthly rainfall deviation from the mean monthly rainfall (black dots) for 18 months prior to harvesting (13 October 2021). (B) The 12-month standardised precipitation index (spi). Rainfall data extracted from *CHIRPS* for site 7 (see Fig. S1 for the monthly deviation at all sites). Monthly rainfall averages and spi were calculated across the period 1981-01-01 to 2022-12-31.

experimental plots (sites 3 and 4; which were ~12-year-old plants planted as part of the Thicket-Wide Plot experiment, *Mills et al., 2015*). However, sites 7 and 8 were within a game farm exposed to herbivores, but all cuttings were harvested from large plants above 1.5 m, thus limiting the range of herbivores those branches were exposed to (*i.e.,* only to Greater kudu, *Tragelaphus strepsiceros*). All cuttings were harvested from large, healthy (free of disease and pest infestation based on visual inspection) individuals. Cuttings were harvested and stored in clear plastic bags and transported back to the laboratory where they were kept at room temperature for four days while being processed. To obtain an estimate of leaf moisture content for each parent-plant, two batches of 30 leaves were removed from each parent-plant on the day of harvesting and placed into sealed containers of known weight. These were then reweighed in the lab to obtain the wet weight, and all leaves were dried at 70 °C for three days before being weighed to determine leaf moisture content; moisture content was calculated as percentage of wet weight. All parent-plants were harvested on the 13th of October 2021. The collection permit was obtained from the

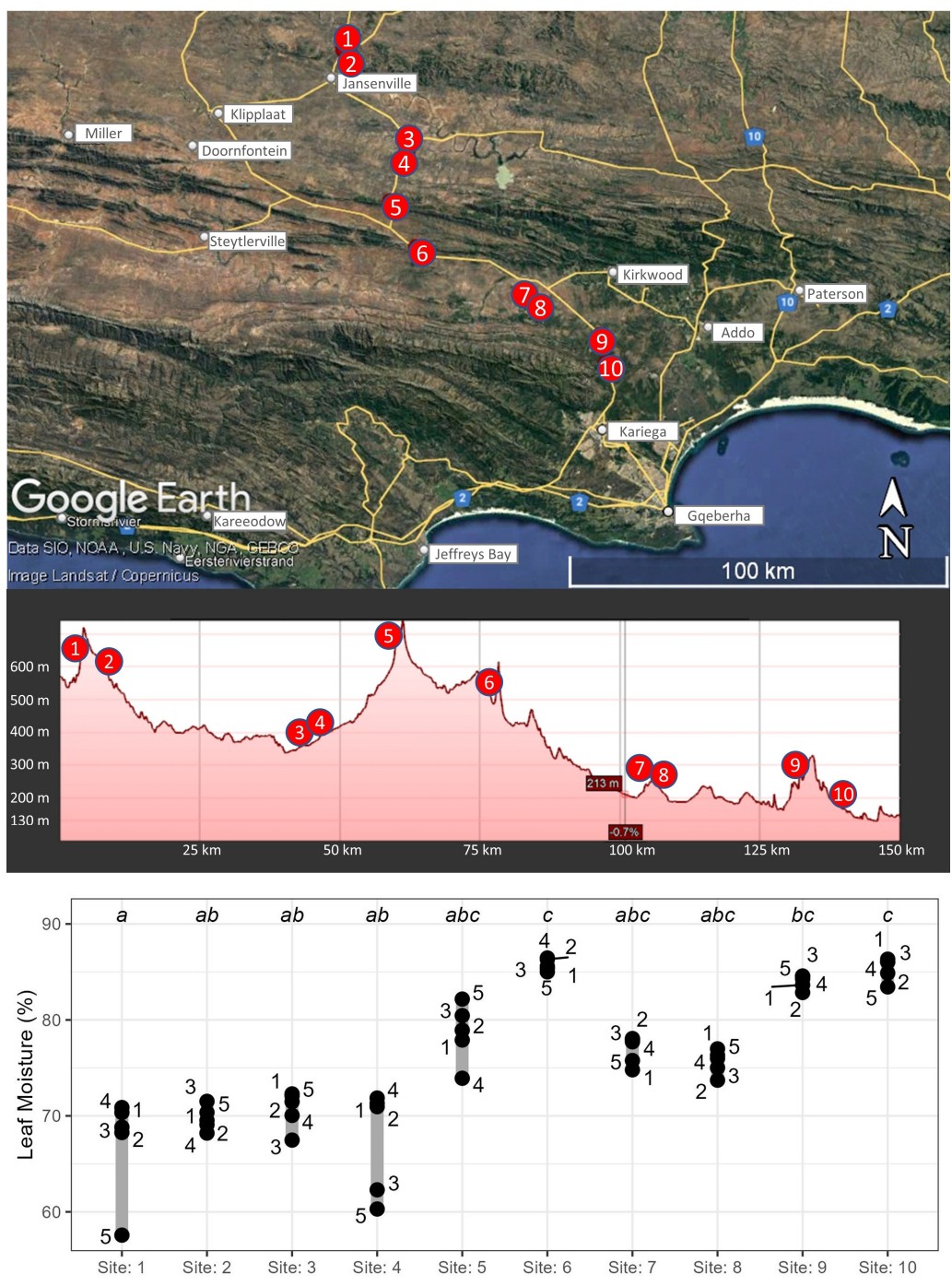

**Figure 2** **Map of harvesting locations and parent-plant leaf-moisture.** (A) A map and elevation profile showing the locations of the ten sites harvested for *Portulacaria afra* cuttings (Map data ©2023 Google). Cuttings were obtained from five parent-plants per site (except site 5, where only four plants were harvested; see text for details). (B) Leaf moisture (percentage of wet weight) of parent-plants harvested at each population; dissimilar superscripts indicate significant differences amongst sites (obtained using the non-parametric Dunn Test of multiple comparisons).

Eastern Cape Department of Economic Development, Environmental Affairs and Tourism (permit number: "HO_RSH_32_2021").

For clarity, throughout the text we use the term "harvest" to indicate obtaining cuttings from parent-plants in-field at the start of the experiment, and "sampling" to denote the cuttings destructively measured from the common garden during the experiment (described below).

## Experiment setup and sampling of cuttings

A common garden experimental design was employed so that we could control the watering regime, variations in soil, and exclude herbivores. Clay-rich soil collected from semi-degraded thicket was used in this experiment. The soil was sieved and mixed to ensure homogeneity, and placed into polypropylene UV-protected plastic seed trays (98-cavity trays with volume of $\sim$90 cm$^3$ per cavity). Cuttings were individually planted into each cavity; in total, 22 trays were used. Long cuttings were trimmed to $\sim$20 cm in length. The mean ($\pm$ standard deviation) stem diameter of the cuttings was 4.99 $\pm$ 0.85 mm and length was 16.9 $\pm$ 1.90 cm. The placement of cuttings from harvesting sites and parent-plants was semi-randomised across the seed trays: cuttings from the same individuals were grouped into batches of seven—the number of cavities per column in the seed tray—and the batches were randomised, in lines, across the trays (*i.e.,* cuttings in columns were randomised); thus, cuttings from a parent-plant were spread across multiple trays. The handling and planting of cuttings into trays took place indoors. After all cuttings were planted, the trays were moved to the common garden, where they were placed in a grid formation, spaced 30 cm apart (see Fig. S2 for photos).

The common garden was an outdoor fenced area located on the Nelson Mandela University campus, Gqeberha, South Africa (Fig. S2), with no obstructions that could shade the trays. Thus, the cuttings were exposed to the same climatic conditions and exposed to full sun, and protected from possible interference from local herbivores. The common garden is located near the coast, and experiences a more benign climate (*i.e.,* lower temperature extremes and higher rainfall) relative to the interior where the parent-plant harvesting sites are located, but still within the natural range of *P. afra*. Nonetheless, the common garden is still within the aseasonal rainfall regime of the region, where phases of drier and wetter periods are common, and *P. afra* is naturally occurring. We watered the trays (to the point of soil saturation) directly after shifting them to the common garden, and thereafter the trays were watered weekly if there were no rainfall events in the previous seven days—we considered this akin to inland field conditions during a wet phase over summer. We did not want soil moisture availability to be a limiting factor in root development in this experiment; dry conditions have been demonstrated to impede root development in *P. afra* (*Galuszynski et al., 2023*). Each week, the trays were moved (prior to any watering) to new, randomly assigned positions within the grid layout.

Destructive sampling of a subset of six cuttings per parent-plant took place across seven separate events, specifically 20, 27, 35, 42, 48, 56, and 103 days after harvesting from the field sites (days 20 to 48 were in November 2021, day 56 in December 2021, and day 103 in January 2022). During each sampling event, the cuttings from each parent-plant were
randomly selected from across all trays. Note that in five instances across the sampling events, the number of cuttings per plant dropped to four due to handling errors.

At each sampling event, the presence or absence of roots was evaluated for the randomly selected subset of cuttings, and roots were also collected for weighing from sampling day 35 onwards. Soil was gently removed from the roots, and the roots were then dried at 70 °C for three days and weighed to four decimal places of a gram.

## Data analyses

The statistical analyses below were conducted in R version 4.2.2, with the *groundhog* library 3.0.0 used for library version control (*Simonsohn & Gruson, 2023*).

The leaf moisture percentage of the parent plants at the time of harvesting was tested for significant differences amongst sites. The leaf moisture was not normally distributed (tested using the Shapiro–Wilk normality test), and thus the non-parametric Kruskal–Wallis test was used. The Dunn's test of multiple comparisons (within each sampling event) was used for post-hoc analysis, with $p$ values adjusted using the Hochberg method. These analyses were conducted using the *shapiro_test()*, *kruskal_test()* and *dunn_test()* functions from the *rstatix* v0.7.2 library (*Kassambara, 2023*). Furthermore, to explore the potential relationship between local rainfall and parent-plant condition, linear regressions were conducted between parent-plant leaf moisture and precipitation summed across varying months (1, 2, 3, 6, 9 and 12 months) at each site (obtained from the *CHIRPS* dataset). To test for a relationship between parent-plant condition and subsequent rooting of cuttings, linear regressions were also conducted between leaf moisture and the mean dry root weight (described below) at the various sampling days. All linear regressions used the *lm* () function in base R.

Root development was measured in two ways: the percentage of rooted cuttings per parent-plant, and the root dry weight. The percentage of rooted cuttings was calculated for each parent-plant at each sampling event (*i.e.,* from 6 cuttings per plant; but this may range from 4 to 7 in some instances). The percentage values were not normally distributed (even after applying transformations such as arcsine, square-root or logit; Shapiro–Wilk normality test), thus the Kruskal-Wallis test was used to detect significant differences amongst sites for each sampling event. Again, Dunn's test of multiple comparisons (within each sampling event) was used for post-hoc analyses ($p$ values adjusted using the Hochberg method).

The mean dry root weight was calculated for cuttings from each parent-plant (for each sampling event) to test whether there were inter-site differences in root development. The mean root weight within sites (per sampling event) was not always normally distributed (Shapiro–Wilk) but did have equal variances (tested using Levene's test). Thus, a slightly different approach was taken. ANOVA analyses were conducted for each sampling event on two sets of data: (1) any sites that had a non-normal distribution were excluded, and (2) all sites. Post-hoc Tukey tests were conducted on both sets of analyses. The Levene's test, ANOVA and Tukey honest significant differences analyses were conducted using the *levene_test()*, *anova_test* () and *tukey_hsd()* functions, respectively, from the *rstatix* library. All figures were made using the following libraries: *ggplot2* v3.4.1 (*Wickham, 2016*),

*ggpmisc* v0.5.2 (*Aphalo, 2023*), *ggConvexHull* v0.1.0 (*Martin, 2017*) and *ggrepel* v0.9.3 (*Slowikowski, 2021*). Dissimilar superscripts were used to indicate significant differences amongst post-hoc pairwise comparisons; these were generated from the *multcompLetters* () function of the *multcompView* ver 0.1-8 library (*Graves, Piepho & Dorai-Raj, 2024*).

Although the basal stem diameter of each cutting was measured prior to planting into each cavity, it was not used as a potential predictor of rooting (percentage or mass) as stem diameter is greatly affected by the moisture content of the stem (which can vary between 55 to 70%).

The descriptions of statistical significance are reported following the conventions suggested by *Muff et al. (2022)*, specifically, $p > 0.1$ is described as no evidence, $p < 0.10 - 0.05$ is described as weak evidence of an effect, $p < 0.05 - 0.01$ is described as moderate evidence of an effect, $p < 0.01 - 0.001$ is described as strong evidence for an effect, and $p < 0.001$ is described as very strong evidence of an effect.

## RESULTS

At the time of *in situ* harvesting of cuttings from the parent-plants, the leaf moisture content varied significantly across sites (Fig. 2B; Kruskal–Wallis, $\chi^2_{(9)} = 44.72$, $p \leq 0.0001$, $n = 50$) and was significantly (strong evidence) and strongly correlated with the amount of rainfall received at each site in the months prior to harvesting (1 to 12 months; $F_{(1,8)} = 5.23 - 16.5$; $p = 0.004$ to $<0.0001$; $r^2 = 0.66 - 0.87$; Fig. S3). Furthermore, leaf moisture content of the parent-plant was significantly (moderate to strong evidence), but weakly, correlated with subsequent cutting root masses at each sampling event (Fig. S4; $p = 0.027$ to $<0.01$; $r^2 = 0.10 - 0.24$).

The results of the Kruskal–Wallis test on the rooting percentage data revealed strong evidence for the effect of site on all sampling events ($p < 0.005$ for all tests; see Table S2). The percentage of rooted cuttings (per plant) increased over time at all sites; however, this trend varied across individuals and sites (Fig. 3). Some sites exhibited high (near 100%) rooting already from day 35 onwards (*e.g.*, sites 6 & 10), whereas some started low and improved rooting over time (*e.g.*, sites 1, 7), and others only had minor improvements but stayed relatively low (<75%, *e.g.*, sites 4 & 8). In a few cases, cuttings from specific parent-plants had consistently low rooting success (*i.e.*, <40%) across all sampling days (Fig. 3).

We also found very strong evidence for the effect of site on rooting within all sampling days for mean root dry weight: $p < 0.0001$ (ANOVA results reported in Table S3); excluding or including sites with non-normal distributions did not affect the ANOVA or post-hoc Tukey results (Table S3 and Fig. S5, respectively). Site 10 had significantly higher root mass than all other sites across all sampling events, while sites 3, 4 and 8 had significantly lower values than many other sites in some sampling events.

Furthermore, parent-plants that exhibited rapid root initiation (*i.e.*, half or more of the cuttings had developed roots 20 days after being harvested from the parent-plants) maintained a higher rooting percentage and, in most instances, higher mean root dry weight up to the end of the experiment (103 days; Figs. 3 and 4). It should be noted that sites 3, 4,

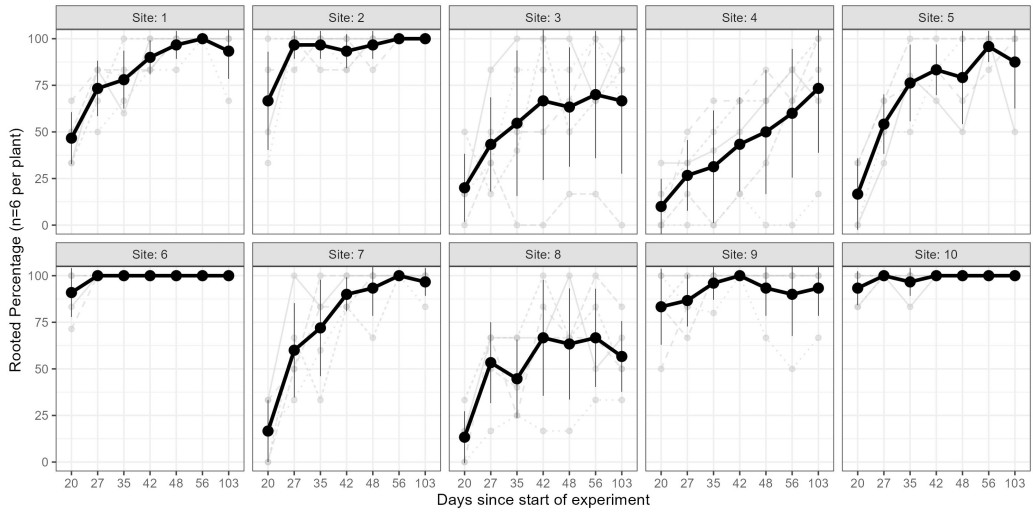

**Figure 3 Rooting percentages across sampling events.** The rooting percentage of cuttings (per parent-plant) by site for each sampling event: mean of all parent-plants (black line; standard deviation shown with thin vertical black lines) and individual parent-plants (grey lines). Cuttings were destructively sampled ($n = 6$ per parent-plant per sampling event). See Table S2 for mean and standard deviation of rooting percentages of the six cuttings per plant, and ANOVA and post-hoc Tukey tests. Sites occurred in three landscape positions: inland mountains (1, 2, 5, 6), inland lowlands (3, 4, 7, 8) and coastal hills (9, 10).

5, and 7, which had relatively low percentages of rooted cuttings after 20 days (10–20%) were able to reach comparatively high proportions of rooted cuttings per individual after the 103-day experimental period (67–97%; Table S2). However, the delayed rooting of these plants resulted in a lower accumulation of dry root weight over the course of the experiment (Fig. 4).

## DISCUSSION

We found strong evidence for differences in root development amongst harvesting sites, both in terms of the percentage rooting and dry root mass. We will discuss these results first in terms of the local environment and its potential influence on parent-plant condition, and then how variable rooting establishment impacts both restoration initiatives and experiments.

The parent-plant material for this study was harvested during a multi-year drought (*Archer et al., 2022*; *Mahlalela et al., 2020*; see Fig. S1). The resulting drought stress likely contributed to variable plant condition across harvesting sites, consequently impacting the root development of the cuttings. Despite the regional drought, moisture availability varied across the study transect (Fig. S1); the coastal hill sites (9 and 10) experienced a greater amount of rainfall than the interior sites (1–8) and the inland mountainous sites (1, 2, 5, and 6) had greater rainfall than neighbouring inland lowlands (3, 4, 7, and 8). This variation in rainfall impacted the leaf moisture content of the parent plants (Fig. 2B, Fig. S3).

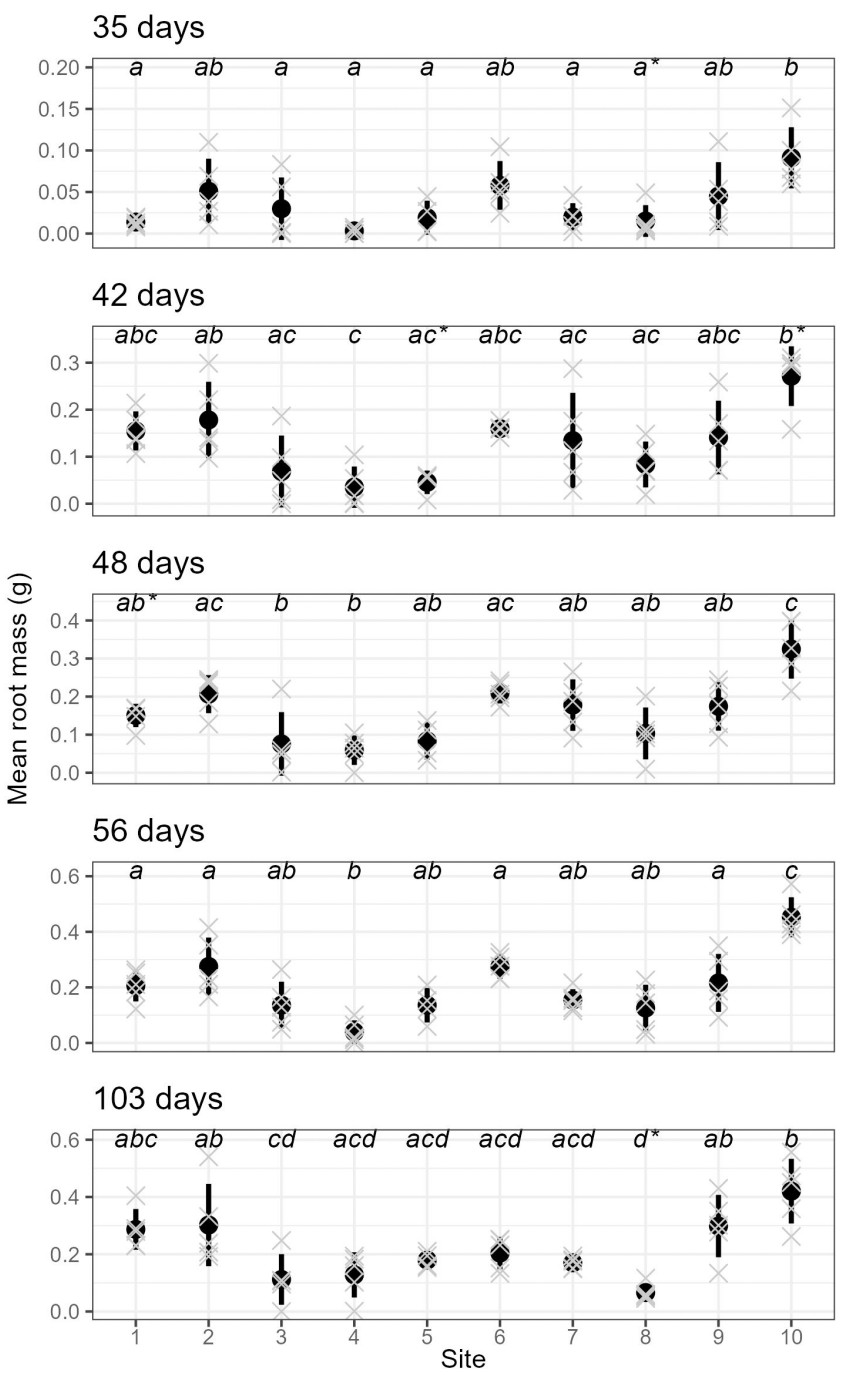

**Figure 4** **The root dry mass of cuttings from different sites across sampling events.** The mean (and standard deviation) root dry mass of cuttings from five separate parent-plants harvested from ten different sites; mean root dry mass is calculated from 4–7 cuttings per parent-plant per sampling event (see text for details; the mean root mass per parent-plant are shown as grey crosses). All sampling events were found to have sites with significantly different root dry mass (see Table S3 for details), and dissimilar letters indicate significant differences amongst sites (note that sites that failed the Shapiro–Wilk normality test with each sampling event are shown using a (*) with the letters).

The physiological consequences of drought stress in *P. afra* include reductions in enzyme activity, decreased leaf chlorophyll content, and a shift towards the dominance of CAM photosynthesis (*Guralnick & Ting, 1987*). Additionally, the relative ratios of stored carbohydrates and soluble sugars in *P. afra* leaves shift in response to seasonal droughts, with the availability of soluble sugars decreasing in drier winter months (*Bews & Vanderplank, 1930*). These physiological changes likely contribute to the ability of *P. afra* cuttings to initiate root development.

The formation of roots in plant cuttings involves the process of cellular redifferentiation, whereby predetermined cells shift their morphogenetic path to an undifferentiated state (mother cells) that can initiate the formation of root primordia (*Husen & Pal, 2007*; *Aeschbacher, Schiefelbein & Benfey, 1994*; *Friedman, Altman & Zamski, 1979*). This process requires an abundance of soluble sugars that provide the energy for protein synthesis and peroxidase activity required for cell division (*Husen & Pal, 2007*; *Bakshi & Husen, 2002*). The decreased availability of soluble sugars reported in drought-stressed *P. afra* (*Bews & Vanderplank, 1930*; or overbrowsed plants, discussed later) may be a contributing factor to the poor and highly variable rooting in plants sourced from sites 3, 4, and 8 (Fig. 3, Tables S2 and S3). Parent-plant leaf moisture content (which was strongly correlated with estimated received rainfall) explains up to 26% of the variation in root mass development of cuttings from the harvesting sites (Fig. S4). This was surprisingly low, largely driven by sites 1 and 2 which had low leaf moisture (~70%) but also fairly good root growth (Fig. 4, Fig. S4). We speculate that low leaf moisture content of a parent-plant does not necessarily have a strong predictive relationship with cutting rooting potential because *P. afra* can remain in a drought-stressed state for a long period of time, switching from C3 to CAM or CAM-idling photosynthesis (*Guralnick & Ting, 1986*; *Guralnick & Ting, 1987*), with no way to determine the time spent in a low leaf moisture state, and thus the overall condition of the plant. Beyond precipitation, the sites harvested in this study represent a variety of environments, including different physical environments (aspect, elevation, and possibly geology), local climate, and browsing pressures. Below we speculate on a range of factors that may also influence parent-plant condition, specifically dew formation, temperature, edaphic conditions, and herbivory.

In addition to topographic effects on rainfall (*i.e.,* mountains *vs* lowlands), dew formation may also be more prevalent in the mountainous areas compared to the lowlands (*Kidron, Yair & Danin, 2000*). Dew is an important source of moisture in arid and semi-arid ecosystems (*Fan et al., 2023*; *Jia, Wang & Wang, 2019*; *Uclés et al., 2014*). Foliar absorption of dew has been demonstrated in a variety of *Crassula* (a genus of CAM succulents commonly found in succulent thicket communities) (*Martin & Von Willert, 2000*). As the quantities of water accumulated during dewfall are relatively low, it is unlikely that it will have long-lasting effects on leaf moisture content, but it may contribute to buffering the physiological responses to drought stress. Thus, foliar dew absorption could potentially explain the high rooting potential of cuttings sourced from mountainous sites with low leaf moisture contents and low rainfall (specifically sites 1 and 2; Figs. S3, S4). However, foliar dew absorption by *P. afra* is yet to be explored.

In addition to affecting local moisture regimes, topography greatly impacts local temperatures. In general, the lowlands in the region experience lower minimum temperatures (*Duker et al., 2015*; *Duker et al., 2020*), which can place more stress on cold-sensitive plant species (*e.g.*, *P. afra*), reducing the available resources for root development (*Fernández et al., 2007*). This may, in part, contribute to the poor rooting observed in lowland sites (3, 4, and 8).

Local edaphic conditions may also impact parent-plant condition by mediating water and nutrient availability (*e.g.*, *Mazaheri & Mahmoodabadi, 2012*; *Bacon, 2009*; *Mamedov et al., 2001*). Thus, soil may influence parent-plant resilience to drought (*Xu et al., 2021*; *Tariq et al., 2017*) and extreme climatic events (*Fernández et al., 2007*). The soil properties along the transect may be highly variable, with rapid and extreme transitions (*e.g.*, bontveld; *Carvalho & Campbell, 2021*). However, establishing the role of soil on parent-plant condition is beyond the scope of this study.

Despite efforts to select *P. afra* plants that were unaffected by local browsing pressure, sites 7 and 8 were located within an area that did include herbivores. This may have contributed to the comparatively low starting leaf moisture content of plants sourced from these sites (Fig. 2B), losing moisture through wounding. Furthermore, the physiological resources required for cellular regeneration (*i.e.*, soluble sugars) may be more limited in these plants as they frequently have to mobilise these resources to respond to herbivore damage. This may have contributed to the variable rates of root development in cuttings sourced from these sites (Figs. 3 and 4). We suspect that moderate to high levels of herbivory will have a pronounced negative effect on parent-plant condition, especially during dry phases, and subsequently impact root development of cuttings.

Thus, multiple factors, or a single factor, may be responsible for the variation in rooting observed in this study. Further research as to the importance of these factors during dry and wet phases is necessary to understand how they impact rooting.

**Parent-plant condition in restoration initiatives and experiments**

The findings presented here (*i.e.*, that source location impacts plant condition and consequently root development in *P. afra* cuttings) have important implications for succulent thicket restoration initiatives. Cutting source location effects have not been considered in *P. afra* planting efforts to date, and as shown here (initially demonstrated in *Galuszynski et al., 2023*) may play an important role in unrooted cutting establishment and survival, particularly under field and nursery conditions. As previously noted, planting of unrooted *P. afra* cuttings is an unreliable restoration practice (*van der Vyver et al., 2021*; *Mills & Robson, 2017*). A biome-wide field experiment—consisting of 330 plots to test the viability of various planting treatments using unrooted *P. afra* cuttings—revealed that cutting survival was affected by stem diameter (*i.e.,* larger cuttings established more readily than small cuttings), uncontrolled herbivory, and planting into the incorrect habitat (*van der Vyver et al., 2021*). However, the reported variability in survival may also be due to the condition of the source populations, which has the potential to override any treatment effects (*Galuszynski et al., 2023*). While cutting size and planting habitat can be easily managed, there is currently no clear method for identifying optimal material

to source cuttings for restoration. As leaf moisture content is not a fully suitable proxy for plant condition, we urge restoration practitioners to focus harvesting efforts towards individual plants that exhibit high leaf moisture content and/or fresh recent growth (typically identifiable by smooth red stems >10 cm long), as this provides some indication that the parent-plant has sufficient resources to support active growth. Rooting these cuttings in a nursery setting before field planting can avoid some of the issues associated with source material effects by filtering cuttings that would fail to initiate root development due to the effect of poor source material.

The results from this study support the findings of *Galuszynski et al. (2023)*, that the source of *P. afra* cuttings can influence experimental results. Thus, we urge experimental work conducted on *P. afra* to account for parent-plant effects and to control for this in experimental design.

## CONCLUSION

Harvesting location had a strong effect on the root development of *P. afra* cuttings grown in a common garden experiment with a watering regime comparable to a wet cycle under field conditions. As rainfall is often unpredictable, and short- and long-term droughts are common in the Eastern Cape, parent-plant condition may have substantially contributed to the highly variable survival rates observed in large-scale restoration and research initiatives using *P. afra*. We find that leaf moisture content, while correlated with root development, had low explanatory value for root development in unrooted *P. afra* cuttings, but may aid in identifying suitable harvesting sites for restoration initiatives under certain conditions. Practitioners will need to account for the effect of local climate variability on plant condition, and develop systems to address this challenge. Furthermore, future research should include site and parent-plant effects in the experimental design.

## ACKNOWLEDGEMENTS

The authors express their gratitude to the reviewers, Abrham Abiyu and an anonymous reviewer, as well as the handling editor, Curtis Daehler, for their insightful and helpful comments during the review process. These insights significantly contributed to enhancing the manuscript.

### Funding

This work was supported by the National Research Fund of South Africa (Grant No. 119379) and the Nelson Mandela Universities' Postdoctoral Research Fellow Grant Program. This work was also supported by the Natural Resource Management programme of the South African Department of Forestry, Fisheries and the Environment (Project No. E1406). The funders had no role in study design, data collection and analysis, decision to publish, or preparation of the manuscript.

## Grant Disclosures
The following grant information was disclosed by the authors:
National Research Fund of South Africa: 119379.
Nelson Mandela Universities' Postdoctoral Research Fellow Grant Program.
Natural Resource Management Programme of the South African Department of Forestry, Fisheries and the Environment: E1406.

## Competing Interests
Alastair J. Potts is an Academic Editor for PeerJ. Robbert Duker is employed by C4 EcoSolutions (Pty) Ltd.

## Author Contributions

- Alastair J. Potts conceived and designed the experiments, performed the experiments, analyzed the data, prepared figures and/or tables, authored or reviewed drafts of the article, and approved the final draft.
- Robbert Duker conceived and designed the experiments, performed the experiments, authored or reviewed drafts of the article, and approved the final draft.
- Kristen L. Hunt performed the experiments, authored or reviewed drafts of the article, and approved the final draft.
- Anize Tempel performed the experiments, authored or reviewed drafts of the article, and approved the final draft.
- Nicholas C. Galuszynski analyzed the data, authored or reviewed drafts of the article, and approved the final draft.

## Field Study Permissions
The following information was supplied relating to field study approvals (i.e., approving body and any reference numbers):

Eastern Cape Department of Economic Development, Environmental Affairs and Tourism: Permit Number: HO/RSH/32/2021.

## Data Availability
The raw data and R code are available in the Supplemental Files.

## Supplemental Information
Supplemental information for this article can be found online at http://dx.doi.org/10.7717/peerj.17471#supplemental-information.

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
