# Peer review of "Restoring South African subtropical succulent thicket using Portulacaria afra: root growth of cuttings differs depending on the harvest site during a drought"

_PeerJ, doi:10.7717/peerj.17471_

## Round 0.1 · original submission · Major Revisions

The reviewers have pointed out various concerns that will need to be addressed in a revised manuscript. From my own reading and based on reviewer feedback, I have two general comments:

1) Additional details are needed regarding methodology and justification for the chosen methodologies.

2) For reasons explained below, I think the wording of the conclusions needs to be revisited.

Methods – I think stem diameter should be used as a factor or covariate in your statistical model. You already measured these values, so it would be simple to add them to the model as a potential explanatory factor.

Planting – the “garden” needs to be described in more detail. In many readers’ minds, a garden is an outdoor area, but you used the word “enclosure” – what does that mean or look like?

Were cuttings watered after planting? Was the soil wet at the time of planting? What type of soil was used? The watering question is especially important since soil water can strongly influence establishment success and root growth, so your experiment is not repeatable without this information. Would cuttings experience a similar water regime during a restoration outplanting project? If not, how are your findings relevant to restoration outplanting?

Figure 1 – Why are these data presented? How were they used? The figure caption refers of sub-figures A) and B) but there is only one figure and no A) or B) shown on the figure.

Figure 2 – I doubt if the tiny light-colored fonts on the Google map and elevation map (x and y axes and top labeling) will be legible if this image is embedded in a formatted publication or web page. Can the authors improve this map by using fonts based on vector graphics rather than grainy pixels, or by using larger fonts or brighter white colored fonts (especially x and y axes of the elevation graph)?

Figure 3 – site 6 and site 10 had the highest leaf water and in figure 3 it is clear that these two sites both also had the highest % rooting and lowest variation in % rooting. This does support a role of leaf water in ensuring high % rooting and ensuring high consistency in % rooting. While reporting statistical patterns is important, visual patterns should be discussed especially when they support a hypothesis. Statistical patterns are a function of sample size and variability within samples across the entire experiment. ANOVA assumes constant variance across treatments, which is obviously false here; P-values alone from ANOVA should not be strongly relied upon to make decisions on the importance of factors / treatments. One could also consider fancier statistical models that assume different variances among treatments / factors).
Figure 3 – How is it possible for the % rooting (n= 6) to fluctuate up and down over time? Is it because one or more plant died? This should be explained in the caption. I think the % rooting curves would be better conveyed as monotonically increasing functions (as you would do for e.g. % seed germination over time).

Figure S2 – looking at the situation after 36 days we see a visual pattern where the four lowest leaf water sites (sites 1-4) tend to have lower root mass than the three highest leaf water sites (sites 6,9,10). The fact that the pattern becomes more obscured over time is not surprising because plants that have established roots will grow based on local conditions in the garden rather than based on their water status more than 30 days earlier. I would have liked to see data on root growth in relation to leaf water within 7-14 days after planting the cuttings. I would expect to see a stronger pattern. Can this be discussed in Discussion?

The authors conclude that “parent-plant identity can have an overriding influence on experimental results”; however, looking at the data (i.e. Figure 4), I am not convinced of an over-riding or even a substantial importance of individual parent plants in determining success. The graphs do not show error bars for parental plant means, but it is essential to see those error bars in order to understand the degree of within-plant versus among-plant variation at each site. I suspect error bars would be overlapping among many parental plants and in most cases, plotting a 95% confidence would show no difference among most pairs of parental plants at a site. I’m not suggesting that there are no cases of parental plants differs, but I don’t think you have made a convincing case that there is a general “overriding” effect of parental plant across this experiment. If you use the strong term “overriding” or something similar to make a generalization of patterns, please provide additional support or justification.

Reviewer 1 ·

Basic reporting

The justification of the study seems to be partly, that P. afra is established poorly in large-scale restoration programs, but it is not clear how this controlled experiment will help to address issues encountered in the field. It would be great if some of the experiments were conducted in the field, possibly at the potential restoration areas (where establishment challenges are encountered), or harvest sites (where the parent-plant conditions are created) for us to isolate the effects of parent-plant conditions on root development, otherwise, it reads more as speculation, than provide solid evidence. This is because the parent-plant conditions might not be the only contributing factor to this poor establishment. There could be many more such as the legacy effects of degradation, weather, soil e.t.c? There seems to be more than one factor affecting the establishment of P. Afra at the restoration sites. It would be nice if some experiments were run at harvest sites and well as the degraded areas where restoration is to take place,

Be consistent in the use of the term root development, which is a collective for percentage rooted cuttings and mean root dry weight. The term is used interchangeably, particularly in the last paragraph of the introduction (Line 102, 103), whereby the objectives are reported collectively as root development. But thereafter and in the results section, the objectives are reported as percentage rooted cutting or root dry mass as opposed to root development. Yes, root development was explained in the abstract as a percentage of rooted cuttings and mean root dry weight, but it would be good to be consistent with the terminology. And what does the environmental gradient in Line 103 refer to? per harvest site or over sampling days, both, per individual, within or across harvest site? Make this clear for each objective, so that it links with how the results were reported. This will make it easier to link each objective to the data set used, analysis, outcomes, and discussion. In other words, specify your objectives to link them to the reported results. Some of the findings reported in the results were not clearly stated as objectives.

Lines 104 – 110 seem to be out of place as they read as methods and results. I suggest that you improve this paragraph by moving lines 104- 110 to the relevant sections. E.g., the description of the harvest sites can be included in the methods section and the findings moved to the result section.

Why are the figure captions above the figures?
Figure 1… label on the actual figure the a and b parts. A figure must be self-explanatory, without looking at the caption and vice-versa.
Figure 2 caption….P. Afra is not italicised

Experimental design

Why was the experiment trialled in a common garden if it is attempting to address problems experienced in the field? Was this to isolate variation caused by the parent plant conditions from field conditions? What informed the choice of the common garden? To what extent did it simulate the field conditions. What kind of soil were the cuttings grown from….where did the soil come from? Harvest site, local soil or what? Line 186?
How old were the other harvest sites Line 151 and what state were there in?
How was the sample size determined? Were 5 plants per site enough to sample from to avoid type I and II errors?

Was spatial autocorrelation accounted for and how considering that “sites encompass a diverse range of environmental conditions, including different elevations (190-635 m)” Line 106, before comparing percentage rooted cuttings across harvesting sites. But since the cuttings were grown in the garden perhaps it does not matter. However, for site 5 and 6, although they are close to each other (spatially), they seem to receive different amounts of rainfall.

Justify the calculation of nested ANOVA using lme?

Why were harvest days not included in the formula structure but they appear in Table S3, Figure 4, how were they incorporated into the model/code which produced Figure 4 ), neither was the interaction of harvest site with harvest day included.

What was the purpose of the climate data harvested using CHIRP?, How does it fit into the analysis? M OR It was simple to show that the harvest sites received different rainfall Line 139 – 141. Meaning we now know that the sites have an influence on the parent-plant conditions.

The model below only specifies one fixed effect/ predictor and it is treating individual plants as random effect. See your R code below.
Line 198 and R code lme(RootMass~Pop,random=~1|Ind,data=.,method="REML")->
model
anova.lme(model,
type="sequential",
adjustSigma = FALSE)->
res
Also, how was the model tested for goodness of fit to justify the fitting of lme model on your data?

How was the data on leaf moisture content analysed? There is a visual Figure S2 and it is reported in the result section but it is not clear from the methods how it was analysed, unless I missed it.

Lower p values do not equate to increased confidence or importance, consider discussing effect sizes to establish the latter.

Validity of the findings

Although your results are compelling, the data analysis should incorporate the predictors shown on the visuals and reported in the results section. For example, Line 212: "Leaf moisture content was found to vary amongst plants within sites and across sites (Figure 2B); however, there was no clear pattern between the leaf moisture content at the time of harvesting and the final mean dry root mass of each parent-plant. However it is not included in the methods, how this data was analysed, but a visual of the analysis was provided in Figure S2.

Additional comments

No comment

·

Basic reporting

Basic reporting
Inmost cases the authors have taken time to create the story in a good language. However, there are some parts of the manuscripts where the authors should consult proficient English speaker. For instance, the method section is difficult to understand.
Specifically the following parts needs authors attention
Lines 36-37
41-42,
91-95
108-110, non-relevant sentence/phrase
136-138
165-172
179-183
227-229
243
524
Lines 103 – 104, there should be evidence for this

Experimental design

Experimental design and results
Line 146: I have strong doubt if five individuals are enough representative of a given site. IT would have been good if more parent plants were included in stead of 42 samples takes from a single plant. It would be good if author present evidence these samples are enough to get properly capture the environmental and individual diversity.
164, common garden ? what ? If it is a common garden experiment, the authors should clear explain the common garden experiment setting.
165 -172, needs re-writing. It is not clear for the reader
171-172, this is un-acceptable/not acceptable reason for not doing proper randomization. The authors should indicate if there are any bottleneck besides handling time.
179 -183, why these days after planting chosen to collect data. Besides, if 6 parent plants are taken eat each session, it will affect the total number of plants remaining at the end of the experiment. Given the sampling was conducted at 7 sessions and 42 parent plants are available.
It would have been good if there were any observation between SPI Vs Soil Moisture Index, foliar moisture content and root initiation.

Validity of the findings

In my view it is valid

Additional comments

It is a well planned and executed experiment. If the authors can give good evidence for some concerns raised in the design section.
The number of references in the reference list are too many. Some of them needs proper citation e.g. line 524-526 , World Agroforestry Centre, Duguma, L., Minang, P., Aynekulu, E., Carsan, S., Nzyoka, J., Bah, A., Jamnadass, R., 2020. From tree planting to tree growing: rethinking ecosystem restoration. through trees. World Agroforestry Centre. https://doi.org/10.5716/WP20001.PDF

---

## Round 0.2 · accepted · Accept

The revised manuscript (including analyses and graphs) carefully addresses the concerns raised by reviewers. I have two minor comments as follows:

L 30 “after 35, 42, 48, 56, and 103 days after” – remove one “after”

L 73 “with no consideration of the condition of the source material used” – you might consider rewording slightly to avoid offense to your hard-working restoration practitioners. I imagine that practitioners do select the healthy-looking cuttings whenever they can be found, while avoiding wasting time on harvesting dead or nearly dead cuttings.